# Thin Cationic Polymer Coatings against Foodborne Infections

Yuliya K. Yushina [1,2,*], Andrey V. Sybachin [3,*], Oksana A. Kuznecova [1], Anastasia A. Semenova [1],
Eteri R. Tolordava [1,2], Vladislava A. Pigareva [3,4], Anastasiya V. Bolshakova [3,5], Vyacheslav M. Misin [6],
Alexey A. Zezin [3,7], Alexander A. Yaroslavov [3], Dagmara S. Bataeva [1], Elena A. Kotenkova [1],
Elena V. Demkina [8] and Maksim D. Reshchikov [1]

1   V.M. Gorbatov Federal Research Center for Food Systems, 109316 Moscow, Russia;
    o.kuznecova@fncps.ru (O.A.K.); a.semenova@fncps.ru (A.A.S.); tolordava.eteri@yandex.ru (E.R.T.);
    d.bataeva@fncps.ru (D.S.B.); e.kotenkova@fncps.ru (E.A.K.); reshchikov@fncps.ru (M.D.R.)
2   N.F. Gamaleya Federal Research Center of Epidemiology and Microbiology, 123098 Moscow, Russia
3   Department of Chemistry, Lomonosov Moscow State University, 119991 Moscow, Russia;
    vla_dislava@mail.ru (V.A.P.); bolshakova@belozersky.msu.ru (A.V.B.); aazezin@yandex.ru (A.A.Z.);
    yaroslav@belozersky.msu.ru (A.A.Y.)
4   A.N. Nesmeyanov Institute of Organoelement Compounds, Russian Academy of Science,
    119334 Moscow, Russia
5   Frumkin Institute of Physical Chemistry and Electrochemistry, Russian Academy of Sciences,
    119071 Moscow, Russia
6   Emanuel Institute of Biochemical Physics, Russian Academy of Sciences, 119334 Moscow, Russia;
    misin@sky.chph.ras.ru
7   Enikolopov Institute of Synthetic Polymeric Materials, Russian Academy of Sciences,
    117393 Moscow, Russia
8   Research Center "Fundamentals of Biotechnology", Russian Academy of Sciences, 119991 Moscow, Russia;
    elenademkina@mail.ru
*   Correspondence: yu.yushina@fncps.ru (Y.K.Y.); sybatchin@mail.ru (A.V.S.)

**Abstract:** Biocidal coatings are known to minimize or terminate development of bacterial and fungicidal infections. In this paper, biocidal activity of seven cationic (co)polymers with amino groups—polyethyleneimine, polyallylamine, polydiallyldimethylammonium chloride/polyhexamethylene guanidine copolymer, diallyldimethylammonium chloride/$SO_2$ copolymer, linear and hyperbranched epichlorohydrin/dimethylamine copolymers, polydiallyldimethylammonium chloride—were tested toward Gram-positive and Gram-negative cells. The polymers showed a significant biocidal effect in both aqueous solution and after formation of polymer films on the hydrophilic glass plates. Polymer films were almost completely removed by water during 10 wash-off cycles, that finally resulted in the ultrathin monolayers with a thickness of several nanometers. A polyethyleneimine film showed the most resistance to water with a 50% loss after three wash-off cycles and 75% loss after six wash-off cycles. Binding and subsequent deactivation of pathogenic microorganisms occurs on the outer surface of cationic polymer films. It is expected that a gradual polymer wash-off will allow renewal of the outer film surface and thereby restore the biocidal properties of the polycationic coatings, including those with a nanoscale thickness.

**Keywords:** cationic (co)polymers; polymer films; nanoscale thickness; stability; biocidal activity

## 1. Introduction

Production, storage and distribution of food products require conditions that minimize or terminate development of infections caused, among other things, by bacterial and fungicidal activity [1–4]. In order to combat pathogenic microorganisms, a wide range of biocidal substances has been suggested that are capable of suppressing their vital activity [5–8]. Conventional low molecular weight biocides have been applied as bacteria-and-fungi killing agents for years [9–11]. However, such biocides often do not have a

long-term effect, show poor adhesion to the surface to be treated, form fragile films which can be easily removed from the surface, etc.

Stable protective films can be fabricated from polymers, some of which demonstrate their own biocidal properties [12–14]. For example, polymers with cationic groups (polycations), dissolved in water, bind to cells and form monolayers of nanometer thickness, which initiate a series of processes, finally leading to a dysfunction of cells and/or their disruption and death [15–18]. After deposition of a polycation aqueous solution over the surface and subsequent drying, a film with bactericidal properties is formed [19–21]. In recent years, progress in the development of antibacterial formulations has been driven by the use of polycation-based nanomaterials and ultrathin coatings down to a thickness of several nanometers [21–26]. Nanostructured biocidal layers can be prepared via alternative electrostatic adsorption of cationic and anionic polyelectrolytes; antimicrobial nanoparticles can be embedded in the layer, thus increasing the biocidal effect of the polymer coating [21,24–27]. A typical thickness of nanostructured coatings does not exceed 100 nanometers.

The literature on biocidic polymers is extensive, with several detailed reviews published in recent years [28–33]. Among polycations the macromolecules with quaternized amino-groups like polydiallyldimethylammonium chloride or quaternized polyethyleneimine (q-PEI) were reported to be the most effective non-specific biocides [30,34]. The charge of these polymers does not depend on pH, so their high antibacterial activity is not affected by the pH of surrounding media. Nevertheless, the PEI with primary and ternary amino groups by itself was also reported to demonstrate antimicrobial activity [17,35]. It is important to stress that some polycations were reported to have low toxicity to humans and the environment [36,37]. Among such polycations there are species that have found their application as flocculants for water purification [38–41].

However, the papers often describe results that are difficult to compare. This is due to the fact that authors and researchers use various polymers—sometimes unique and synthesized for a specific research, different types of microorganisms and experimental conditions—polymer solutions or suspensions or films, from nanometers to hundreds of microns in thickness, cell cultivation protocols and cell concentration, methods for assessment of cell survival, temperature, etc. In addition, the activity of polymeric biocides should be correlated with physico-chemical properties of polymer films. Only in this case can one find a formulation with optimal biocidal and operational characteristics.

In the current article, we describe seven commercial polymers with cationic units responsible for biocidal activity. In the control experiments, the seven polycations were tested against bacterial cultures in aqueous solutions. Then, the thin polycation coatings were prepared on the glass slides, and their biocidal activity were tested again. Special attention was paid to quantifying the resistance of polycation coatings to wash-off with water. This aspect is of key importance for fabricating long-term anti-microbial films on various surfaces including those in industrial and public premises. We show that the polymer monolayer, a few nanometers in thickness, retained on the glass surface even after multiple wash-off procedures. The results allowed us to select polycations from the group of seven commercially available polymers for preparing the films, down to nanometer thickness, with an optimal "biocidity vs. water resistance" relationship.

## 2. Materials and Methods

### 2.1. Materials

The following polymers were used in the research. Polyethyleneimine, polyallylamine, polydiallyldimethylammonium chloride, and polyhexamethyleneguanidine were purchased from Sigma-Aldrich (St. Louis, MO, USA); copolymer of dialyldimethylammonium chloride and $SO_2$ was purchased from Technolog (Sterlitamak, Russia); linear copolymer of dimethylamine and epichlorohydrin was purchased from BSC Chemicals (Sterlitamak, Russia); hyperbranched copolymer of dimethylamine and epichlorohydrin crosslinked with diaminomethane was purchased from SNF-East (St. Petersburg, Russia). The

seven cationic samples were prepared from individual polycations or their mixture (CPs). For the details, see Supplementary Information Table S1. The samples carried permanent positive charges (CP4, CP5, CP6 and CP7) or acquired positive charges via acidification of aqueous polymer solutions (CP1 and CP2); one polymer sample (CP3) contained both permanent and acid-produced positive charges. Also, the structures of polycations varied from linear (CP2, CP3, CP4, CP5, CP6) to branched (CP1 and CP7). The degrees of polymerization of polycations varied from oligomeric for CP7 to long-chained for CP1. The detailed analysis of non-trivial polymers'—flocculants CP4, CP5 and CP7—structures were published elsewhere [38,39,42].

*Pseudomonas aeruginosa* and *Listeria monocytogenes* samples were taken at a food enterprise in the Moscow region in September 2020 and identified with the use of standard microbiological testing.

### 2.2. Methods

The Otto method adapted for screening studies (the flowing drop method) was used in this work. A 1.5% meat-peptone melted agar was poured into sterile Petri dishes, then cooled and dried for 30 min at 30 °C. *P. aeruginosa* was added to the TSB broth at a concentration of $1 \times 10^6$ CFU/mL. An overnight TSB broth culture was 100-times diluted and 50 μL of the diluted culture was put in the Petri dish closer to its rim. The dish was tilted, and the bacterial culture flowed to the opposite dish side. After 15 min, a 30 μL drop of an aqueous polymer solution was deposited on the runoff trajectory of the bacterial culture drop. The sample was dried and incubated for 24 h at 37 °C. An interruption of the "bacterial pathway growth" by the deposited polymer was considered a positive antimicrobial result.

*Pseudomonas aeruginosa* bacteria were stained for the cytometric tests using Thermo's LIVE/DEAD BacLight Bacterial Viability and Counting Kit (Waltham, MA, USA). The following ingredients were added to 987 mL of 0.9 wt% aqueous sodium chloride solution: 1.5 mL of prepared SYTO 9 solution, 1.5 mL of prepared propidium iodide solution, and 10 mL of the bacterial culture. After carefully stirring, the mixture was allowed to incubate in light-protective Eppendorf tubes for 15 minutes. Using a Guava EasyCyte flow cytometer (Merk-Millipore, Darmstadt, Germany), live and dead cell populations were quantified, with live cells turning green and dead cells turning red.

*Pseudomonas aeruginosa* and *Listeria monocytogenes* daily cell cultures were diluted 100 times with a nutritional medium for the microbiological evaluation of bacterial viability in solution, and the polymers were then added. In TSB broth, a daily cell culture was produced by culturing microorganisms at a concentration of $1 \times 10^6$ CFU/mL. The mixtures were incubated for 18 h at 37 °C, diluted 10 times with sterile deionized water, and then subjected to a standard methodology for counting colony-forming units (CFU). As controls, bacterial cultures devoid of polymers were employed.

Glass slides were washed with potassium bichromate/sulfuric acid mixture, potassium hydroxide/methanol mixture, bidistilled water, and finally air dried at room temperature in order to make polymer films. The sample was then sprayed with 200 L of a 2 weight percent aqueous cationic polymer solution to a freshly cleaned glass slide and air dried at room temperature to produce a 0.15 mm thick polymer film. The broth cell culture was placed in the glasses with the deposition of polymer films, and the cells were then incubated for 18 h at 37 °C. The glasses were then transferred to test tubes containing saline solution and vigorously shaken after being cleaned three times with distilled water. CFUs were identified in the resultant washes using a common procedure.

To visualize living and dead bacteria, the polymer-modified glasses with deposited cells were washed three times with distilled water, stained with the Live/Dead Kit as described above and identified with microscope: green color for living cells and red color for the dead.

The weight loss of the CP films was analyzed by the procedure described elsewhere [43]. A 2.25 cm$^2$ freshly cleaned glass slide was weighed using VLA-120 M Gosmetr

precise balances (St. Petersburg, Russia). Then, 200 μL of the 2 wt% aqueous solution of CP was deposited on a glass slide so that the entire surface was covered with the solution. The sample was left to dry in air overnight, then weighted again and the weight of polymer film was calculated as the difference between the weight of the CP-covered glass slide and the weight of the initial glass slide without polymer. In each washing cycle, 200 μL of water was applied to the CP-covered glass, which completely covered the polymer coating. After 2 min, the liquid was eliminated and the sample was left to dry in air overnight and weighted. The loss of weight was calculated as a difference between the weights of the sample before and after the washing cycle. Ten successive washing cycles were performed.

Using a scanning probe microscope, the Nanoscope IIIa (Santa Barbara, CA, USA), in tapping mode in air, atomic-force microscopy (AFM) imaging was carried out. We used silicon cantilevers from TipsNano (Moscow, Zelenograd, Russia) with resonance frequencies between 140 and 150 kHz. A 15 mm × 15 mm cover glass was submerged for 5 min in a 1 weight percent polycation solution. The excess polymer was then removed from the glass by rinsing it in DI water for one minute, after which the sample was allowed to dry in the air.

For determination of the minimum inhibitory concentration (MIC), *P. aeruginosa* and *L. monocytogenes* strains were studied. Overnight cultures of bacteria were diluted with BHI broth to obtain a standardized inoculum of $5 \times 10^7$ CFU/mL. Further, the resulting inoculums were diluted with fresh rich BHI medium and placed in the wells of a 96-well polypropylene microplate with the addition of various concentrations of polymers (from 0.5 mg/mL to 20 mg/mL) to obtain a final cell density of $5 \times 10^5$ cells/mL and the total content volume was 150 μL. Aliquots of aqueous solutions of polymers were prepared by successive two-fold dilutions of stock solutions and the addition of equal volumes of diluted polymers to each well. Diluted cell suspensions without the addition of polymers served as controls for each culture. The range of final concentrations of polymers in the wells was from 0.5 to 20 mg/mL. For each polymer concentration, the inhibitory effect was tested in triplicate.

The minimum bactericidal concentration (MBC) was defined as the lowest concentration of each of the tested polymers that results in the destruction of 99.9% of the tested bacteria [44].

The MBC was taken to be the highest concentration of the polymer at which bacteria did not grow on the agar plates. To do this, cultures in microplates were grown for 4 h and a series of inoculations was made from wells with no visual growth. For seeding, 10 μL was taken from the wells corresponding to MIC and higher concentrations of MIC, sown by spatula on PCA agar medium, and incubated for 24 h at 30 °C.

## 3. Results and Discussion

The polymers were tested for their biocidic activity in the same conditions as described in the experimental part.

Bacterial cells, Gram-positive *L. monocytogenes* and Gram-negative *P. aeruginosa*, both known as causative agents of foodborne infections and opportunistic infections [45,46], were isolated from objects of a meat processing plant and a poultry plant. This allowed working with microorganisms circulating in the production environment and forming stable consortia on the production surfaces.

Figure 1 demonstrates the biocidal activity of cationic polymers in an experiment known as a flowing drop method. A drop of cell culture, *P. aeruginosa*, in a nutritious broth was applied on a solid nutrient medium in a Petri dish, and the drop was allowed to drain off. Then, a drop of an aqueous polycation solution was applied over a "bacterial culture line". After incubation of the Petri dish for 18 h, a growth of the cell culture was detected along its runoff trajectory but interrupted at the sites of polymer deposition.

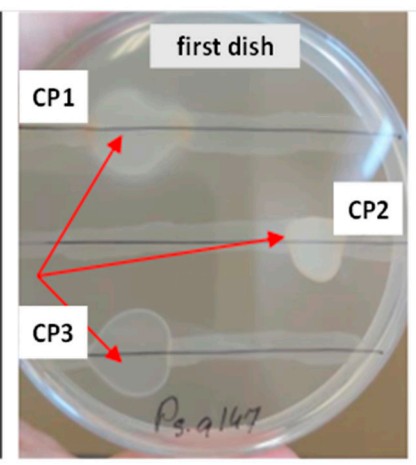 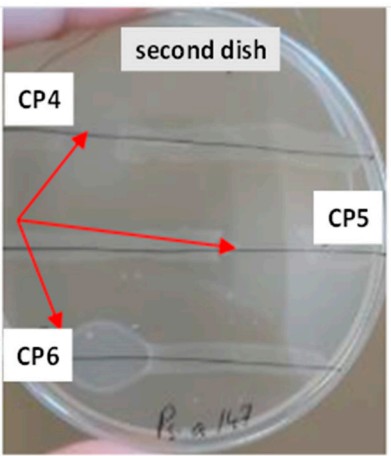 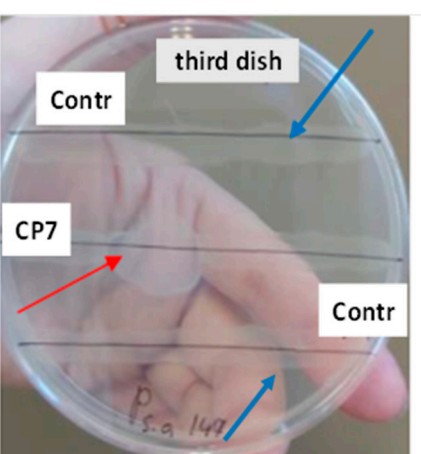

**Figure 1.** Photos of Petri dishes after successive deposition of *P. aeruginosa* culture and aqueous polymer solutions. First dish: CP1, CP2 and CP3 (red arrows); second dish: CP4, CP5 and CP6 (red arrows); third dish: CP7 (red arrow) and control (blue arrows). See details in text.

Figure 1 shows typical photos of Petri dishes after successive deposition of a *P. aeruginosa* culture and aqueous polycation solutions. In all photos, there are spots, which are located on the trajectories of the cell culture flow. These spots appeared in the places where drops of polymer solutions were applied (marked with red arrows). It is clearly seen that deposition of four polymers: CP3, CP4, CP5 and CP6, interrupted the bacterial growth. Three other polymers: CP1, CP2 and CP7, also demonstrated the antibacterial activity, which however was less pronounced because of milky white color of the interrupting spot. Thus, all seven polycations suppressed the growth of *P. aeruginosa*.

The next step was to quantify the biocidal activity of polycations using two methods, flow cytometry and microbiology. The first is based on an ability of SYTO 9 dye to stain living cells in green and dead cell in red (from yellow to dark brown). Table 1 shows the number of living *P. aeruginosa* cells after 5 h (as described in the experimental part) incubation of the cell–polymer suspensions. In the positive control (with no polymers), the concentration of living cells before polymer addition was of $2.6 \times 10^7$ mL$^{-1}$. All cationic polymers showed a high antimicrobial activity, killing more than 99% of the bacterial cells. The toxic effect of cationic polymers in solutions is realized due to the high conformational mobility of macromolecules, which provides "adjustment" of the cationic groups of polymers to the attacked cells, followed by reorganization of the cell membrane and violation of cell functioning.

**Table 1.** Cytometry method for testing antimicrobial activity of polymers toward *P. aeruginosa* in solutions.

| Sample | Control | CP1 | CP2 | CP3 | CP4 | CP5 | CP6 | CP7 |
|---|---|---|---|---|---|---|---|---|
| CFU/mL | $2.6 \times 10^7$ | $2.0 \times 10^4$ | $2.0 \times 10^4$ | $6.0 \times 10^4$ | $7.0 \times 10^4$ | $4.8 \times 10^5$ | $4.0 \times 10^4$ | $2.0 \times 10^4$ |

Another method for testing the biocidal activity of polycations is the microbiological assessment of bacterial survival. The results for two types of bacteria, *L. monocytogenes* and *P. aeruginosa*, are summarized in Table 2.

**Table 2.** Microbiological method for testing activity of polymers toward *L. monocytogenes* in solutions.

| Sample | | Control | CP1 | CP2 | CP3 | CP4 | CP5 | CP6 | CP7 |
|---|---|---|---|---|---|---|---|---|---|
| CFU/mL | *L. monocytogenes* | $5.0 \times 10^7$ | 0 | 0 | 0 | $3.0 \times 10^6$ | $1.0 \times 10^6$ | 0 | 0 |
| | *P. aeruginosa* | $8.0 \times 10^7$ | 0 | 0 | $3.3 \times 10^4$ | $1.7 \times 10^3$ | $2.0 \times 10^4$ | $2.3 \times 10^5$ | 0 |

Three polymers, CP1, CP2 and CP7, showed the absolute antibacterial effect, killing 100% of both *L. monocytogenes* and *P. aeruginosa*. Two polymers, CP3 and CP6, were 100% active against *L. monocytogenes* and deactivated the overwhelming majority (>99%) of *P. aeruginosa*. The last two polymers, CP4 and CP5, killed more than 95% of *L. monocytogenes* and more than 99% of *P. aeruginosa*. In other words, either cationic polymer in an aqueous solution killed at least 95% of bacteria common in food processing plants.

Last but not least, the bactericidal qualities of films created by drying polycation aqueous solutions were investigated. Glass slide pieces were covered in polymer layers before being inoculated with *L. monocytogenes*. According to Table 3 statistics, all polymers effectively inhibited the growth of bacterial cells. These findings are in good agreement with those of other researchers who investigated the biocidal properties of multilayered coverings made by alternate electrostatic adsorption of cationic and anionic polyelectrolytes with a cationic polymer in the outer layer exposed to water (see, for instance, [21]). Such coatings ranged in thickness from 5 to 10 nanometers [22].

**Table 3.** Inhibition of *L. monocytogenes* bacterial film formation by polymers.

| Sample | Control | CP1 | CP2 | CP3 | CP4 | CP5 | CP6 | CP7 |
|---|---|---|---|---|---|---|---|---|
| CFU/mL | $4.0 \times 10^7$ | 0 | 0 | 0 | 0 | 0 | 0 | 0 |

The Live/Dead Kit was used to visualize the cells on the glass surfaces. The fluorescence of cells treated with the Live/Dead Kit and seen under a fluorescent microscope is depicted in Figure 2. Propidium iodide penetrated dead bacteria through flaws in the cell walls and painted the bacteria from pale yellow to dark brown. SYTO-9 from the kit exclusively dyed living bacteria green. The *L. monocytogenes* cells adhered to a control glass with no polymer (Figure 2, picture 1), which exhibits a vivid green tint that unmistakably shows the intact structure of the adsorbed cells. In contrast, images 2–8 of cells on glasses coated with cationic polymers show a wide range of hues, from yellow to extremely brown, demonstrating that the cells died after depositing on the films. Thus, the results of fluorescence investigation of the polymer films (Figure 2) are in good agreement with the microbiological data shown in Table 3.

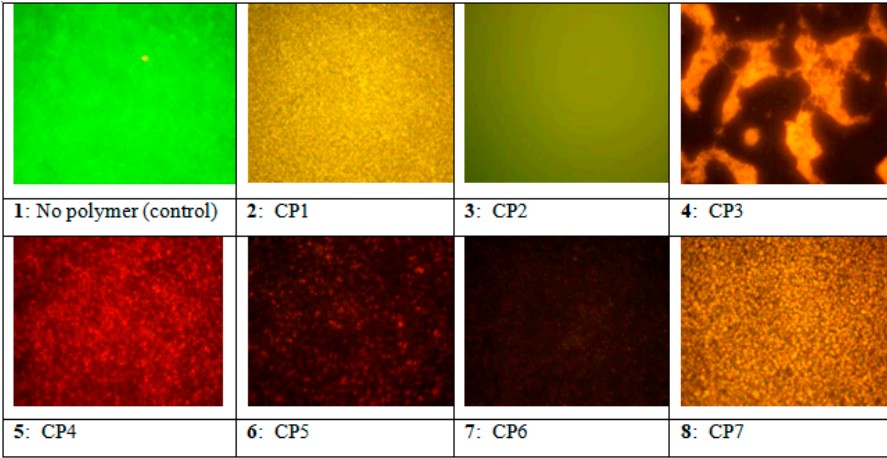

**Figure 2.** Fluorescent microscope images of *L. monocytogenes* cells attached to the glass surfaces with polycations. The cells were treated with the Live/Dead Biofilm Viability Kit. Adapted from [47].

During the experiment, it was reliably shown that all the studied polymers had different inhibitory and antimicrobial activities. While the details surrounding antibacterial activity of polycations is still challenging, it is believed that such macromolecules are capable of disrupting the bacterial cell membrane, resulting in death of bacteria cells [48]. In the first stage, the electrostatic adsorption of polycations on bacterial membrane takes place.

Then, the formation of an interfacial complex leads to destabilization of the membrane. Polycations violate the integrity of the cell membrane allowing them to enter into the cell and bind to intracellular molecules such as DNA and proteins, leading to further cell damage and eventually cell death.

The most potent growth inhibitory effect at the lowest exposure concentration was polymer CP7 (see Table 4). The concentration of 1 mg/mL turned out to be the minimum inhibitor, at which visual growth was not detected on the liquid medium, and also the minimum bactericidal, at which less than 99.9% of cells grew on agar plates. Such an effect was observed for a Gram-positive culture of *L. monocytogenes*. On a Gram-negative culture of *P. aeruginosa*, polymer CP7 acted similarly at a concentration of 5 mg/mL. Different antimicrobial activity of the CP6 polymer was observed for Gram-positive and Gram-negative bacteria. A large concentration variation (5 mg/mL and >20 mg/mL) was found to maximize cell culture growth inhibition (MCG) of *P. aeruginosa* and *L. Monocytogenes*. For polymer CP4, the same minimum bactericidal concentrations were shown for both Gram-positive and Gram-negative cultures. This makes the CP 4 polymer universal in relation to the above objects.

**Table 4.** MIC and MBC methods for testing activity of polymers toward *L. monocytogenes* and *P. aeruginosa* in solutions.

| Polymers | P. aeruginosa | | L. monocytogenes | |
|---|---|---|---|---|
| | MIC, mg/mL | MBC, mg/mL | MIC, mg/mL | MBC, mg/mL |
| CP1 | 10 | 20 | 5 | 5 |
| CP2 | 10 | 10 | 10 | 20 |
| CP3 | 20 | 20 | 5 | 10 |
| CP4 | 10 | 20 | 20 | 20 |
| CP5 | 10 | 10 | 20 | 20 |
| CP6 | >20 | >20 | 2.5 | 5 |
| CP7 | 5 | 5 | 1 | 1 |

A key question concerns the stability of cationic polymer films, especially in rooms with a higher humidity and/or subjected to wet processing. In order to consider the question, four polycationic polymers from the initial seven were taken: CP1 with a set of primary, secondary and ternary amino groups; CP6 with quaternary amino groups; as well as CP5 and CP7, linear and hyperbranched dimethylamine-epichlorohydrin copolymers. All these CPs belong to different groups of cationic polymers and demonstrated pronounced biocidal properties (see above). Also, the choice of these CPs for the testing of coatings was determined by the fact that they are approved as environmentally friendly, possess low toxicity in contact with humans, and are commercially available [49].

Modification of the surfaces with a polycation-based coating could be achieved by covalent or non-covalent binding [50–53]. The first approach usually requires chemical grafting of the polycationic chains to surface or polymerization of cationic monomer units on the functionalized surface, or by grafting-through approach [54]. The second approach requires non-covalent (Coulomb forces, dipole–dipole interactions, etc.) binding of the functional groups of the surface and functional groups of polymers. The simple substrate that allows one to implement non-covalent electrostatically driven adsorption of cationic macromolecules is glass surface with anionic silanol groups [55].

Two methods of immobilization of polycations were used. In the first, the glass plates with slightly acidic silanol groups on the surface were dipped in a 1% aqueous polymer solution for 5 min. After that, the glass plates were taken out of solution and put into a vessel with deionized water that resulted in a removal of a polymer excess. Thus prepared glass plates were dried and examined with an AFM technique (Figure 3). This technique is traditionally used for fabricating thin polymer films with a thickness down to a few nanometer. In our case, the thickness of the polymer layer varied from polymer to polymer but always remained within a nanometer interval that definitely showed a formation of

a monolayer of cationic polymer macromolecules electrostatically attached to the anionic glass surface.

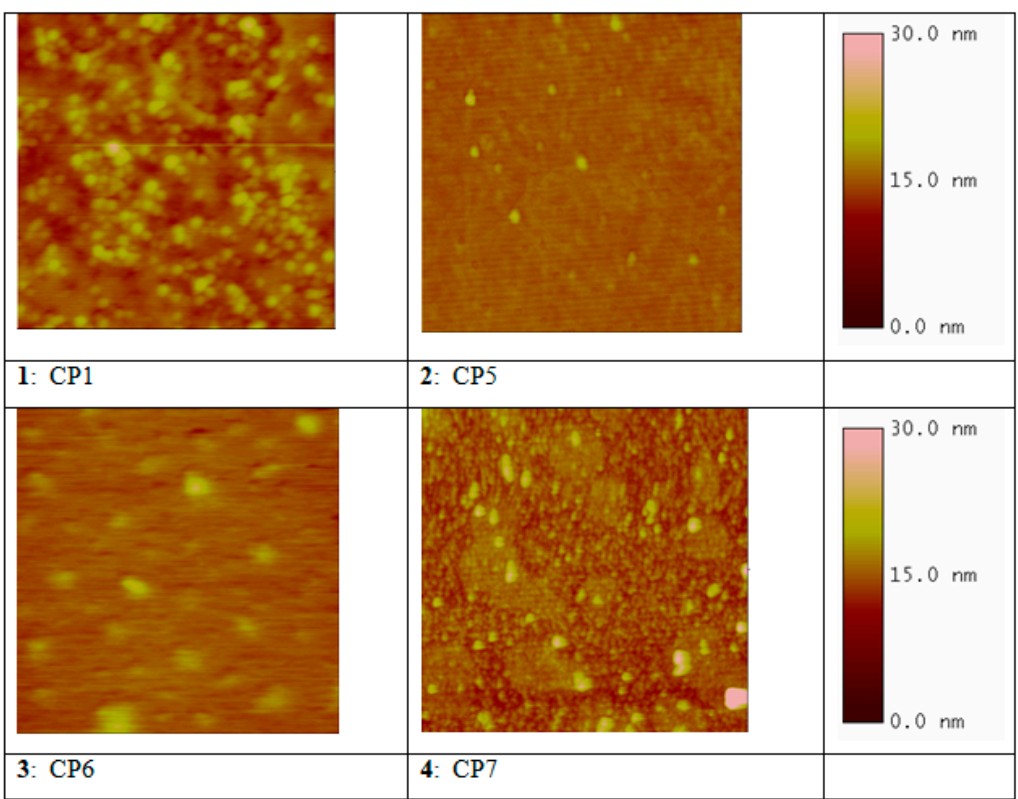

**Figure 3.** AFM images of four polycation layers on the glass surface (2000 nm × 2000 nm scanning area).

The monolayer of polycations has a weak potential as a protective biocidal coating. The monolayer with a nanometer thickness works until its surface has free space for biological particles that come from the air. In a polycation multi-layer in which the thickness multiple exceeded the nanometer mark, the outer surface ensured binding and deactivation of microorganisms as well. However, in this case the polymer film can be cleaned via partial dissolution in water that will result in removal of a thin polymer layer with deactivated pathogenic cells. After dead cells are removed, the cationic surface can re-adsorb and neutralize new living cells. It can be expected that a successive repetition of the wash-off procedure will finally lead to the formation of a polymer layer with a few nanometers in thickness.

In accordance with the second "multi-layer" approach, the polycationic films were prepared via deposition of 2 wt% CP aqueous solutions on glass plates followed by drying the samples in air to constant weight. It is this procedure that was used in order to fabricate films for the antibacterial experiments (see above). All four polymers formed continuous transparent films with good adhesion to the glass substrate and a thickness of 0.15 mm that was evidence of multi-layer film formation (for additional information on the film imaging see Supplementary Information Figure S1). The polycations from films were partially removed with water (see details in the experimental part), 10 successive wash-off cycles were performed. Figure 4 reflects a loss of CP weight vs. the number of wash-off cycles.

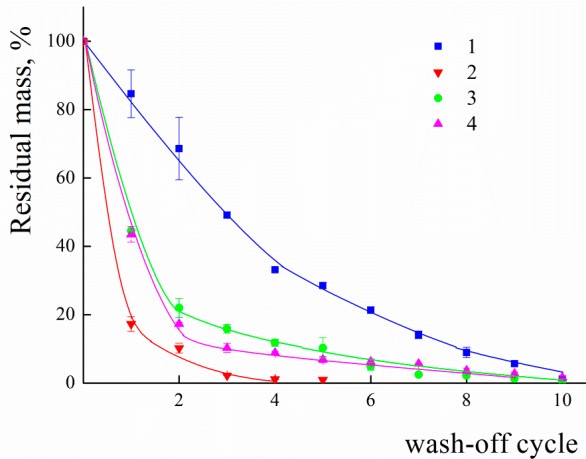

**Figure 4.** Weight loss of CP films vs. the number of wash-off cycles (see details in the text). CP1 (1), CP5 (2), CP6 (3) and CP7 (4).

As follows from the figure, all films almost completely lost the major share of the adsorbed polymers in the course of the wash-off procedure: the CP5 film after four wash-off cycles (curve 2), three other films from CP1, CP6 and CP7 after 10 wash-off cycles (curves 1, 3 and 4). The CP1 film was the most resistant with a 50% polymer loss after three wash-off cycles and a 75% loss after six wash-off cycles (curve 1). A slightly higher stability of the CP1 films can be attributed to hydrophobic interactions of hydrocarbon groups in the polymeric chains. Thus, the wash-off procedure resulted in removal of the vast majority of the deposited polymers. However, the nanoscale polycation layers remained on the anionic glass surface after ultimate flushing.

## 4. Conclusions

Polycations are known to be effective disinfectants [56–59]. In this research, we attempted to analyze the role of chemical nature and structure of the polycations on their biocidal activity.

All seven cationic polymers (CPs), used in the study, showed a pronounced activity against Gram-positive and Gram-negative cells, known as causative agents of foodborne infections and opportunistic infections. The biocidal effect of polymers was manifested in both aqueous solution and after formation of polymer films on the hydrophilic glass plates. The CP toxicity was due to cationic groups of polymers capable of binding to the negatively charged surface of cells and inducing structural rearrangements in the cell membrane. No recommendation about the choice of chemical nature and polymer chain structure of polyelectrolyte to ensure the ultimate antimicrobial effect could be suggested due to the fact that most of the polycations under investigation demonstrated effective biocidal activity toward Gram-negative and Gram-positive bacteria.

Four of the seven polymers, namely CP1, CP5, CP6 and CP7, were tested for their ability to form stable films on the glass. All of them gave multi-layer films on the glass surface, 0.15 mm in thickness. The CP1 (polyethyleneimine) film demonstrated the most resistance to water. It is the outer surface of the cationic polymer films that ensures binding and deactivation of microorganisms coming from the air. As shown above, repeated washing procedures progressively reduced the thickness of the polymer films. However, the newly obtained films obviously retained a positive charge on their surface, which, in accordance with the generally accepted model, should have a biocidal effect on adsorbed cells. An ultimate wash-off resulted in the nanoscale layers of biocidal coatings electrostatically attached to the glass substrate.

The formation of the biocidal coatings from individual polycations should be considered from several points of view. On the one hand, the possibility to renew the biocidal surface with wash-off of the upper layer with water could benefit compared with fixed

coatings formed by grafted biocidal macromolecules. On the other hand, vanishing of the polymer layer could result in change of physical and mechanical properties of the coatings. Also, hydrophilic macromolecules have poor adhesion to hydrophobic surfaces. So, individual polycations should be considered as effective biocidal components of the system that will ensure the formation of the protective coatings on the surfaces with different hydrophilic/hydrophobic balance. In order to construct such systems, our findings allow one to choose the optimal polycation. This choice should be driven by cost-effectiveness, technology of the formation, and physical and mechanical properties of the resulting coatings, but not the structure of the polycation because, for most of the polycations, the antibacterial activity was almost independent of the chemical nature of cationic units.

**Supplementary Materials:** The following supporting information can be downloaded at: https://www.mdpi.com/article/10.3390/coatings13081389/s1, Table S1: structure formulas of polyelectrolytes; Figure S1: image of CP1 film on the glass substrate after formation of the coating (a) and after wash-off (b). The scale bar is 150 μm.

**Author Contributions:** O.A.K., project administration; A.A.S., resources; Y.K.Y., methodology; E.R.T., V.A.P., A.V.B. and A.V.S., investigation and writing—original draft preparation; D.S.B., V.M.M. and E.A.K., investigation; A.A.Z., writing; A.A.Y., conceptualization; E.V.D. and M.D.R., formal analysis. All authors have read and agreed to the published version of the manuscript.

**Funding:** This study was conducted on the state assignment of the V.M. Gorbatov Federal Research Center for Food Systems of the Russian Academy of Sciences, scientific number No. FNEN–2019–0007 (in part of the methodology for assessing antimicrobial properties of substances).

**Institutional Review Board Statement:** Not applicable.

**Informed Consent Statement:** Not applicable.

**Data Availability Statement:** Data is contained within the article or supplementary material.

**Conflicts of Interest:** The authors declare no conflict of interest.

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
