# Peer review of "Thin Cationic Polymer Coatings against Foodborne Infections"

_coatings, doi:10.3390/coatings13081389_

Round 1

Reviewer 1 Report

·         The introduction should provide a clearer background on the significance and relevance of biocidal coatings in various applications. It should highlight the existing challenges in preventing bacterial and fungal infections and how these coatings can address those challenges.

·         It would be helpful to include a brief explanation of the different types of polymers tested in this study, such as their structures and properties. This will aid readers who are not familiar with these specific polymers.

·         Please provide more details about the experimental methods used to evaluate the biocidal activity of the polymers. Include information on the concentrations, exposure times, and specific test organisms used in the experiments.

·         The results section should include quantitative data and statistical analysis to support the claims made about the biocidal effect of the polymers. This will enhance the reliability and credibility of the findings.

·         Consider providing a comparison between the biocidal activity of the different polymers tested. This will help readers understand the relative effectiveness of each polymer and their potential practical applications.

·         The mechanism of action underlying the biocidal effect of the polymers should be discussed in more detail. Elaborate on how the cationic nature of the polymers interacts with the bacterial and fungal cells, leading to their deactivation.

·         Discuss the implications of the wash-off cycles on the long-term stability and effectiveness of the polymer films. Are there any limitations or potential challenges associated with the gradual wash-off process?

·         Include a discussion on the potential toxicity or side effects of the polymers, especially in applications where they come into contact with humans or the environment. Address any safety concerns associated with the use of these biocidal coatings.

·         Provide a comprehensive analysis of the advantages and disadvantages of using the described polymers compared to other biocidal coatings or antimicrobial agents. This will help readers understand the unique features and potential drawbacks of this particular approach.

·         Consider discussing the potential applications and industries where these biocidal coatings can be utilized. Are there any specific areas or materials where these coatings would be particularly beneficial?

·         If available, provide information on the cost-effectiveness of the polymers and their potential scalability for large-scale production. This would be valuable for practical implementation and commercialization considerations.

·         Address the stability and durability of the polymer films over time. Do they retain their biocidal activity after prolonged exposure to environmental conditions or repeated wash-off cycles?

·         Consider expanding the discussion on the experimental limitations of the study. Are there any factors that may have influenced the results or could be improved in future research?

·         Include additional references to support the claims and findings presented in the paper. This will help readers contextualize the study within the broader scientific literature.

·         Clarify any technical terms or abbreviations used throughout the paper. Ensure that readers from diverse backgrounds can understand the content without confusion.

·         Include more details about the characterization techniques used to analyze the polymer films, such as surface morphology, thickness measurements, or chemical composition analysis. This will provide a more comprehensive understanding of the film properties.

·         Consider including images or figures of the polymer films to visually illustrate their formation and wash-off process. This will enhance the clarity and visual appeal of the paper.

·         Provide a concluding section that summarizes the main findings and their implications. Also, discuss potential future research directions and areas for further investigation based on the outcomes of this study.

·         Proofread the manuscript carefully to eliminate any grammatical or typographical errors. A well-written paper will enhance its readability and overall impact.

·         Overall, the paper provides valuable insights into the biocidal activity of cationic polymers, but it would benefit from addressing the suggested improvements mentioned above. Revise and refine the manuscript accordingly to enhance its scientific quality and contribution to the field. 

Overall, the language used in the paper is clear and concise.

Pay attention to sentence structure to ensure clarity and readability.

Check for consistent use of verb tense throughout the paper.

Use appropriate punctuation marks to enhance the flow of the text.

Author Response

-The introduction, discussion and conclusion sections were expanded.

-In this manuscript we tried to analyze number of polycations with different characteristics. Polymers differ in molecular weight, amino-group nature, linear/branched structures and so on. As a result there was found no significant impact of these factors on biocidal activity of these polymers. While PEI, PDADMAC polyhexamethyleneguanidine and polyallylamine are trivial macromolecules, the technical flocculants are not widely discussed in scientific literature. We have expanded the text with description of differences in molecular structures with corresponding references.

-The mechanism of the biocidal action of the polycations is still under discussion. The antibacterial activity of polycations is primarily due to their ability to disrupt the bacterial cell membrane and cause cell death. Polycations are believed to interact with bacterial membranes through electrostatic interactions, forming a complex with the bacterial cell wall or membrane, leading to membrane destabilization. Once the bacterial membrane is disrupted, polycations can enter the bacterial cell and bind to intracellular molecules such as DNA and proteins, leading to further cell damage and eventually cell death. The following fragment was added to the text

-Biocide action of adsorbed polycationic film could be divided into two separate ways. First, during wash-off the desorbed polycation is capable to interact with bacteria in water suspension providing biocide action towards non-adsorbed microorganisms. Second, the residual polycation in adsorbed layer ensures biocidal action against adsorbed microorganisms.

With the increase of the number of the wash-off cycles the antibacterial action of the desorbed polycation should decrease as the number of desorbed molecules from the film will be lower. The efficiency of biocide action of the upper layer of the remained film will be the same. However, the adsorbed bacteria cells will finally form layer of dead cells that will screen new bacteria from the macromolecules. Thus, the wash procedure is also necessary for the renewing of the surface of protective polycationic layer. Without renewing of the surface one could expect gradual lowering of biocidal action of the coating.

-The presented polycations are used as flocculatns, as components of desinfectants in dental surgery and so on. So, they have been proved to have low toxicity for the humans. We have also tested PEI and PDADMAC for the toxicity towards animals with the result supporting the exist data on their safety. However, we have not examined toxicity for the all of the samples. So, we did not include this information in the manuscript.

-This is very important task. But to provide the complete compare of the different coatings the largescale work should be done using different control samples. So, we consider it as individual task that was out of the scope of the current research.

-The polymers under investigation were studied as potential components for biocide coatings against foodborn infections. In order to modify the surfaces of different nature (glass, plastics etc) the of hydrophilic polymers should be modified with hydrophobic units to ensure better adhesion and resistance to desorption (the example could be fined in Pigareva et al. Polymers 2022, 14, 1247 10.3390/polym14061247). . So, the current polymers unlikely to be used as individual-component system to form the protective coating.

-Up to date the price of these polymers vary in range of 1000-2000 $ per 1000 kg. The most expensive polycations are PEI and polyallylamine as they are not mass-produced polymers. As we find in our investigation the effective biocidal activity could be achieved using most of the polycations under investigation. So, the cheap commercially available flocculants could be used as biocidal components.

-In this research we did not focus on this question. However, our experimental data on PEI and PDADMAC activity after formation of the fresh coating and after intensive washing allows us to state that the coatings retain their antibacterial activity.

-As it was mentioned above, we do not consider these polycations as final biocide compositions for the creation of protective coatings. The further modification of polyelectrolytes is required to create the effective coating. The idea of this very research was to analyze the polycations available on the market and to find the marcomolecules with the highest biocidal properties. Our findings allow us to use most of the polycations without restrictions in further experiments targeting to create the effective antibacterial coating.

-The introduction, discussion and conclusion sections were expanded.

-The formation of the continuous films was confirmed using the optical microscopy. The typical image was added to supporting information. The measurement of surface profile and thickness was made with AFM. According to the limitations of the experimental methods the initial thickness of the initial films was measured by optical microscopy and the thickness of residual films was measured with AFM.

-The suggested images were added to the Supporting information.

Reviewer 2 Report

This paper describes the antibacterial and antifungal activities of amino group-containing cationic polymers as coatings. Though this is not the first time these phenomena have been demonstrated with such polymers, the current study not only exhibited their effectiveness but also proved their long-term resilience under practical situations. all the studies have been conducted carefully. However, I will recommend expanding the introduction section as for a general reader it was not written very well to emphasize the importance of the presented work. Also, some editing of English is required in various places (mainly grammatical issues). Other than these, the paper seems scientifically sound the conclusions are supported by relevant experiments. Therefore, I recommend its publication.

 Some editing of English is required in various places (mainly grammatical issues

Author Response

The introduction, discussion and conclusion sections were expanded

Reviewer 3 Report

The reviewer found the idea of the submitted manuscript titled ‘Thin cationic polymer coatings against foodborne infections interesting and can be published in Coatings. However, authors might consider the following for the improvements of the submitted article:

1.      Authors have mentioned that ‘The results allowed to select polycations from the group of seven commercially available polymers for preparing the films, down to nanometer thickness, with an optimal “biocidity vs. water resistance” relationship.”, I am just curious about the previously done studies of this comparison. Please mention 1 or 2 citations related to this kind of report.

2.      In my opinion, Table 1 should not be the complementary part of this manuscript. I am not forcing you to remove it, However, I want you to find some optimal way if possible.

3.      Can authors mention the time span information for wash off-cycle?

Author Response

-The corresponding references were added to the text.

-The Table was moved to Supplementary information

-The time for each cycle was 2 min. The residual layer of polycations remained even after 10 cycles (controlled by AFM)